# Resting Energy Expenditure Is Elevated in Asthma

**DOI:** 10.3390/nu13041065

**Published:** 2021-03-25

**Authors:** Jacob T. Mey, Brittany Matuska, Laura Peterson, Patrick Wyszynski, Michelle Koo, Jacqueline Sharp, Emily Pennington, Stephanie McCarroll, Sarah Micklewright, Peng Zhang, Mark Aronica, Kristin K. Hoddy, Catherine M. Champagne, Steven B. Heymsfield, Suzy A. A. Comhair, John P. Kirwan, Serpil C. Erzurum, Anny Mulya

**Affiliations:** 1Pennington Biomedical Research Center, Baton Rouge, LA 70808, USA; Jacob.Mey@pbrc.edu (J.T.M.); Kristin.Hoddy@pbrc.edu (K.K.H.); Catherine.Champagne@pbrc.edu (C.M.C.); Steven.Heymsfield@pbrc.edu (S.B.H.); John.Kirwan@pbrc.edu (J.P.K.); 2Inflammation and Immunity, Lerner Research Institute, Cleveland Clinic, Cleveland, OH 44195, USA; matuskb@ccf.org (B.M.); petersl@ccf.org (L.P.); wyszynp@ccf.org (P.W.); koom@ccf.org (M.K.); sharpj1@ccf.org (J.S.); aronicm@ccf.org (M.A.); comhais@ccf.org (S.A.A.C.); erzurus@ccf.org (S.C.E.); 3Respiratory Institute, Cleveland Clinic, Cleveland, OH 44195, USA; pennine@ccf.org (E.P.); mccarrs@ccf.org (S.M.); mickles@ccf.org (S.M.); zhangp@ccf.org (P.Z.)

**Keywords:** resting energy expenditure, body composition, nutrition, respiratory function, airway inflammation

## Abstract

**Background:** Asthma physiology affects respiratory function and inflammation, factors that may contribute to elevated resting energy expenditure (REE) and altered body composition. **Objective:** We hypothesized that asthma would present with elevated REE compared to weight-matched healthy controls. **Methods:** Adults with asthma (*n* = 41) and healthy controls (*n* = 20) underwent indirect calorimetry to measure REE, dual-energy X-ray absorptiometry (DEXA) to measure body composition, and 3-day diet records. Clinical assessments included spirometry, fractional exhaled nitric oxide (FENO), and a complete blood count. **Results:** Asthmatics had greater REE than controls amounting to an increase of ~100 kcals/day, even though body mass index (BMI) and body composition were similar between groups. Inclusion of asthma status and FENO in validated REE prediction equations led to improved estimates. Further, asthmatics had higher white blood cell (control vs. asthma (mean ± SD): 4.7 ± 1.1 vs. 5.9 ± 1.6, *p* < 0.01) and neutrophil (2.8 ± 0.9 vs. 3.6 ± 1.4, *p* = 0.02) counts that correlated with REE (both *p* < 0.01). Interestingly, despite higher REE, asthmatics reported consuming fewer calories (25.1 ± 7.5 vs. 20.3 ± 6.0 kcals/kg/day, *p* < 0.01) and carbohydrates than controls. **Conclusion:** REE is elevated in adults with mild asthma, suggesting there is an association between REE and the pathophysiology of asthma.

## 1. Introduction

Asthma is a syndrome of airway inflammation and airflow obstruction and is one of the most common chronic diseases in the United States [1]. Obesity is a risk factor for asthma [2], but in many cases asthma precedes obesity, indicating that asthma is also a risk factor for obesity [3]. While genetic and environmental factors play a role in both obesity and asthma, recent studies hint that changes in underlying bioenergetics may also be a connecting factor. Airway smooth muscle and epithelial cells and platelets from asthmatics have increased mitochondrial respiration [4,5], while murine models of asthma also show changes in bioenergetics [4,6,7]. Evidence from both preclinical models [8] and children [9,10] suggests that asthma increases resting energy expenditure (REE), potentially contributing to a negative energy balance and delayed or impaired growth [11,12]. However, whether REE is elevated in adults with asthma and/or is related to respiratory function and inflammation remains to be investigated.

A higher body mass index (BMI) and body fat have been associated with asthma [2,13,14]. Thus, the concept of increased REE in asthmatics seems to juxtapose this long-standing asthma–obesity association, as increases in REE in isolation contribute to a more negative energy balance (i.e., weight loss). However, in asthmatic children with increased REE, compensatory increases in dietary intake have been reported to yield similar net energy balance to controls [10]. Further, energy balance is impacted by complex physiology, including body fat distribution [15], immunological reactions [16], and dietary intake [17], but this has not been completely addressed in asthmatics.

Here, we hypothesized that REE is increased in adults with asthma, and the increase is related to increased inflammation, as measured by circulating white blood cell count and fractional exhaled nitric oxide (FENO), a standardized clinical assessment of inflammation in asthma [18]. To test this hypothesis, we measured REE, body composition, fat distribution, and habitual dietary intake, as well as lung function, FENO, and circulating white blood cells and types, in asthmatics and healthy controls.

## 2. Materials and Methods

### 2.1. Study Design

This cross-sectional study on energy expenditure and body composition in adults with asthma and healthy controls was conducted at the Cleveland Clinic, Cleveland, OH. Participants were enrolled as a part of the Asthma Inflammation Research Translational Program (clinicaltrials.gov, NCT01536522). Participants were recruited by public advertisement in the Greater Cleveland area between August 2018 and June 2019. This study was approved by the Institutional Review Board Committee at the Cleveland Clinic (IRB Protocol #10-1049, approval date 22 March 2018 by Bridget Howard, Esq., CIP, the Executive Director of Cleveland Clinic IRB and Human Research Protections).

### 2.2. Participants

Participants were screened based on age (>18 years), BMI (<40 kg/m^2^), and a medical history. Participants with asthma were included if they had 1) asthma diagnosed by a medical specialist and/or 2) a history of a positive methacholine test and/or reversibility of >10% of forced expiratory volume in 1 s (FEV_1_). Control participants were included if they had no history of lung disease. Additionally, participants were excluded for diabetes, active smoking status or smoking history of greater than 10 pack-years, or any other significant respiratory disease, cardiac disease, acute or chronic renal failure, or the presence of clinically relevant diseases or comorbidities. All participants provided written informed consent before initiation of study procedures.

### 2.3. Clinical Assessments and Blood Chemistry

Prior to the study procedures, participants were instructed to fast for 10–12 h and withhold their medications for 24 h. Fasting blood samples were collected, processed, and analyzed by the Clinical Research Unit at the Cleveland Clinic. Airway inflammation was evaluated using a standardized single-breath FENO test (NIOXVERO, Circassia, Oxford, UK). Spirometry was performed using an electric spirometer (OxyCon Pro, Carefusion Healthcare, San Diego, CA, USA) according to American Thoracic Society guidelines [19]. Forced vital capacity (FVC) and forced expiratory volume in 1 s (FEV_1_) and the FEV_1_/FVC ratio were reported. Baseline and bronchodilator-treated spirometry were repeated until three acceptable measures were obtained. The average of three measures is reported. Bronchodilator reversibility was assessed by the maximal achievable FEV_1_ after bronchodilator administration compared to baseline.

### 2.4. Body Composition

Height (cm) and body weight (kg) were measured by standard procedures to one-tenth of a unit. Body composition was quantified using a three-compartment model (bone mineral, fat mass, fat-free mass) by dual energy X-ray absorptiometry (Lunar iDXA, GE Healthcare, Chicago, IL, USA). Body fat distribution (android and gynoid regions) was determined using pre-set analytical regions (ENCORE version 13.60, GE Healthcare).

### 2.5. Resting Energy Expenditure (REE)

Indirect calorimetry was performed in the fasted state with the participant laying supine as previously described [20,21]. Briefly, expired air was continuously sampled for 20 min using an automated system (Vmax Encore, Viasys SensorMedics, Yorba Linda, CA, USA) in a semi-darkened, thermoneutral (22 ± 1 °C) environment under a ventilated hood. A steady-state of 10 min was selected for analysis [22]. The effect of asthma on REE was assessed by comparing indirect calorimetry-measured REE (REE_IC_) to four validated REE prediction equations (Mifflin–St. Jeor, REE_MSJ_ [23]; Harris–Benedict, REE_HB_ [24]; World Health Organization equations, REE_WHO_ [25]; Oxford equations, REE_OX_ [26]; Appendix A). The differences between predicted REE_MSJ/HB/WHO/OX_ and measured REE_IC_ (residuals) were calculated and compared between asthma and control and presented as percentage-predicted REE. A cohort-specific REE equation was created by multiple linear regression modeling using four previously defined criteria (age, sex, fat mass, and fat-free mass) [27] according to best practice [28]. The model was then used to assess the effect of asthma status, lung functions, measures of systemic inflammation (white blood cell counts), and airway inflammation (FENO) on REE.

### 2.6. Dietary Analysis

Habitual dietary intake was assessed using 3-day dietary records. Participants were instructed on how to document all food and beverages consumed for 3 non-consecutive days, including 2 weekdays and 1 weekend day. Dietary analysis was conducted by the Dietary Analysis Core at Pennington Biomedical Research Center using the Food and Nutrient Database for Dietary Studies (FNDDS) (USDA 2015–2016). The Healthy Eating Index was calculated to assess how well a participant’s diet reflected recommendations in the 2015–2020 Dietary Guidelines for Americans [29].

### 2.7. Statistical Analysis

Statistical analyses were performed using PRISM 7 (GraphPad Software, La Jolla, CA, USA) and Excel (Microsoft Corporation, Seattle, WA, USA). Data were assessed for normality according to the Shapiro–Wilk test (normality was accepted at *p* > 0.05). Differences between groups (asthma vs. control) were assessed by an unpaired Student’s *t*-test for normally distributed data or Mann–Whitney U test for nonparametric data. Pearson (parametric) and Spearman (nonparametric) correlations were used to investigate relationships between REE, body composition, immune cells, and dietary intake with respiratory function. For multiple linear regression modeling, increases in adjusted *R*^2^ were used to assess model improvement. Data are expressed as mean ± SD. Statistical significance was accepted at *p* < 0.05.

## 3. Results

### 3.1. Characteristics of Participants

Asthmatics and controls were similar in age (Table 1). The asthma group contained a greater percentage of females (asthma: 76%; control: 55%), which is consistent with the female-predominant demographics of adults with asthma [30]. Asthmatics showed a trend toward a lower percentage of predicted FVC (*p* = 0.05) and FEV_1_ (*p* = 0.08), but FENO was similar among asthmatics and controls. Additional clinical characteristics, asthma exacerbations, and list of medications are available in Appendix A, respectively. Asthmatics had a higher incidence of nasal polyp (*p* = 0.01), bronchitis history (*p* = 0.04), and allergy (*p* = 0.04) than healthy controls. We did not find any significant differences in smoking, pneumonia, breathing sleep disorder, and emphysema between the two groups. An asthma control test (ACT) of asthmatic patients showed that only 29% had an ACT score less than 19 in our population. To control their asthma symptoms, about 58% and 37% of our asthmatic population took inhaled corticosteroids and leukotriene modifiers, respectively. Less than 10% of our population had taken biologics to manage severe asthma symptoms. These data suggest an overall mild to moderate, well-controlled asthma phenotype.

### 3.2. Body Composition

Table 2 shows the body composition analysis measured by dual-energy X-ray absorptiometry (DEXA). Asthmatics trended toward a higher BMI (*p* = 0.06). Bone mineral density and total lean tissue were similar between groups. Total body fat (kg) (*p* = 0.02), body fat percentage (*p* = 0.03), and gynoid fat distribution (*p* = 0.03) were higher in asthmatics. Female asthmatics had a higher body weight and BMI (both *p* = 0.02) than female controls, and this was accompanied by greater body fat (*p* = 0.04), fat-free mass (*p* = 0.05), and fat free mass index (*p* = 0.03).

BMI correlated inversely with FVC (percentage-predicted, *r* = −0.28, *p* = 0.03; Figure 1A) and positively with FENO (*r* = 0.32, *p* = 0.01; Figure 1B). The android-to-gynoid (A/G) ratio trended toward an inverse correlation with FEV_1_ (percentage-predicted, *r* = −0.25, *p* = 0.05; Figure 1C) but was positively correlated with FENO (*r* = 0.45, *p* < 0.01; Figure 1D).

### 3.3. Resting Energy Expenditure (REE) and Respiratory Quotient (RQ)

Resting energy expenditure by indirect calorimetry (REE_IC_) predicted REE_MSJ/HB/WHO/OX_, and residuals (differences between REE_IC_ and predicted REEs) are presented in Table 3. REE_IC_ was similar among asthmatics and controls, but REE is a function of body size and varies by gender. To take into account gender and body size, a comparison of residuals between the measured REE_IC_ and the REE_MSJ/HB/WHO/OX_ prediction equations revealed that asthmatics consistently had a higher REE compared to controls (REE_MSJ_: 108.9 kcals/day, *p* = 0.02; REE_HB_: 111.3 kcals/day, *p* = 0.01; REE_WHO_: 106.4 kcals/day, *p* = 0.06; REE_OX_: 97.1 kcals/day, *p* = 0.07; mean REE increase: 105.9 ± 318.9 kcals/day, *p* = 0.03). When assessed as percentage of predicted REE, similar to how we compare lung function as percentages of predicted, asthmatics again had a significantly greater percentage predicted REE than controls (Control vs. Asthma: REE_MSJ_: 96 ± 14% vs. 103 ± 14%, *p* = 0.01; REE_HB_: 91 ± 14% vs. 97 ± 13%, *p* = 0.02; REE_WHO_: 91 ± 14% vs. 98 ± 13%, *p* = 0.05; REE_OX_: 95 ± 14% vs. 101 ± 14%, *p* = 0.06; Figure 2A). The REE (%-predicted_MSJ_) correlated with both white blood cells (*r* = 0.34, *p* < 0.01, Figure 2B) and neutrophils (*r* = 0.32, *p* = 0.01, Figure 2C), and these correlations were in agreement for REE_HB_, REE_WHO_, and REE_OX_ equations. The addition of asthma status or FENO to the REE_MSJ/HB/WHO/OX_ prediction equations improved each model for estimating REE in asthma. Data are presented in Table 4. Next, to estimate resting energy expenditure in asthmatic subjects that can be applied in clinical care, a cohort-specific REE equation was developed from multiple linear regression modeling. Data are presented in Table 5, and detail of summary statistics are presented in Appendix A. Model 1 was established using previously known criteria (age (years), sex (male = 1, female = 0), fat mass (kg), and fat-free mass (FFM) (kg)), according to best practice in the field [31,32]. Model 1 was improved by adding asthma status as shown (Model 2, asthma diagnosis = 1, no asthma diagnosis = 0), which suggests an additional contribution of 73.9 kcals/d (Mean (95% CI); 73.9 (−62.9, 210.6)) for individuals with asthma. Model 3 was established by adding the FENO value to Model 1 and improved the adjusted *R*^2^ from 0.552 to 0.563. Addition of white blood cell (WBC) or neutrophil counts (in millions of cells) to age, sex, fat mass, and FFM (Model 1) were presented in Models 4 and 5, respectively. Both WBC and neutrophil cell counts’ addition improved Model 1 independently. No measures of lung function (FEV_1_% predicted, FVC% predicted, FEV_1_/FVC ratio) improved the REE model beyond what was expected by adding additional variables. Details of statistics are shown in Appendix A, Model 6, 7, and 8, respectively. Additionally, we explored the effect of age of asthma onset (Appendix A), asthma exacerbations (Appendix A), inhaled steroid (Appendix A), short-acting beta agonist medication (Appendix A), or leukotriene modifiers on REE in the asthmatic population. We found no differences in REE for these subgroups.

Respiratory quotient was similar between groups and was inversely associated with the FEV_1_/FVC ratio (*r* = −049, *p* < 0.01; Figure 3A) and positively associated with maximum bronchodilator reversibility (*r* = 0.37, *p* = 0.02; Figure 3B) in asthmatics.

### 3.4. Immune Cells and Inflammation

Asthmatics had a higher white blood cell count (WBC; 26%, *p* < 0.01), primarily driven by elevations in neutrophils (millions of cells, 29%, *p* = 0.02) (Table 6).

### 3.5. Habitual Dietary Intake

Asthmatics reported consuming fewer calories (−19%, *p* < 0.01) and less carbohydrates (−28%, *p* < 0.01), compared to controls (Table 7). Both groups scored similarly on the Healthy Eating Index (HEI-2015) score, which measures adherence to the 2015–2020 Dietary Guidelines for Americans, but individuals with asthma reported consuming less sugar per day (26 g, *p* < 0.01).

## 4. Discussion

### 4.1. Main Findings

REE is responsible for ~70% of total daily energy expenditure and represents the energy spent at rest to maintain whole body homeostasis, including respiration, body temperature regulation, and immune response. Herein, we find that individuals with asthma have higher REE than healthy control individuals and that the addition of asthma status to validated REE prediction equations improves REE prediction accuracy.

### 4.2. Elevation of REE in Asthma

Few studies have investigated the effect of asthma on REE. Zeitlin and colleagues studied children with asthma and controls (mean age = 8.4 years) and showed greater absolute REE in children with asthma [33]. Benedetti and colleagues compared REE in three groups of adolescents (mean age = 12 years): (1) overweight with asthma, (2) lean with asthma, and (3) overweight without asthma. They reported REE/kg was higher in lean children with asthma but not in overweight asthmatic children as compared to their control counterparts [9]. Maffeis and colleagues investigated only male children with and without asthma (mean age = 9 years) and normalized the data to fat-free mass, reporting greater REE/kgFFM in children with asthma. Dietary record data suggest that children with asthma compensate for the increase in normalized REE by consuming more calories, yielding no net energy deficit compared to controls [10]. A strength of these studies is the well-matched age, height, and weight of study participants. Still, it is difficult to draw a conclusion on the effect of asthma on energy expenditure due to the young age of the study populations, the predominance of males in the studies, and the lack of standardized assessments of REE between reports (e.g., absolute REE, REE normalized to body mass (REE/kg), or REE normalized to FFM (REE/kgFFM)). Further, there are significant methodological flaws to utilizing absolute REE or ratio-based normalization of REE to assess differences between groups [31,32,34].

In our study, the control and asthma groups had a mean BMI that placed them in the overweight/obese category, and both groups had similar REE based on indirect calorimetry (REE_IC_). Standard REE equations have been shown to overestimate actual REE in overweight populations [26,35,36,37]. To take into account age, sex, and body composition, which all affect REE, we determined REE using predicted equations. The REE residuals provide the essential data to estimate the fit of prediction equations to REE_IC_. The finding of lower absolute residuals in asthma is in agreement with the literature that REE prediction equations overestimate REE in overweight/obesity and that the increase in REE in the asthma group acts as a corrective factor, making the prediction equations more accurate in the overweight/obese asthma group. However, the use of REE residual data in this context includes both the direction and magnitude of the residuals, the difference of which quantifies the impact of asthma status on REE. This difference between asthmatics and controls becomes particularly apparent when represented as a percentage-predicted REE. This was true no matter which standard prediction equation was used, i.e., Mifflin–St. Jeor, Harris–Benedict, WHO, and Oxford. These findings led us to explore a cohort-specific multiple linear regression model for more accurate REE prediction in asthma. Addition of asthma status, airway inflammation (FENO), or blood cell count to REE prediction improved the REE prediction model. The magnitude of the impact of asthma on REE was similar between the residual comparisons (~6.6% of REE), percentage-predicted REE (~6.5%), and the linear regression modeling (~4.7% of REE). Overall, this suggests REE is increased in adults with asthma. The increase was 73.9 kcals/day or ~4.7% by linear regression modeling and 105.9 kcals/day or ~6.6% by prediction equation residuals. Our results align with evidence from preclinical models [8] and prior studies in asthmatic children [9,10,33]. The findings concur with well-established studies of elevated REE observed in other airway obstructive lung diseases, such as chronic obstructive pulmonary disease (COPD) [38,39]. Additional studies are warranted to corroborate these findings, whether conducted in the clinical setting using REE-prediction equations compared to REE_IC_ or a more intensive modeling approach using DEXA-measured body composition.

### 4.3. Body Composition

Increased body weight and adiposity have been associated with more exacerbations of asthma and hospitalizations [15]. In agreement, we observed relationships between elevated BMI, body fat percentage, and an android-type fat distribution pattern with reduced measures of respiratory function or FENO. This is in contrast to prior reports suggesting that individuals with asthma have increased body fat and a gynoid-type fat distribution [40,41]. However, we observed no group differences when accounting for sex and BMI. Taken together, our data suggest individuals with asthma do not inherently have higher body fat or a different body fat distribution compared to non-asthma controls. Nevertheless, independent of asthma status, both body fat and an android-type fat distribution were related to reduced respiratory function. Fat accumulation can contribute mechanically to lung function via mechanisms of a reduction in respiratory system compliance, reduction in lung volume, reduction in respiratory muscle strength, increase in airway resistance, and increase in pulmonary diffusion [42]. Additionaly, adipose tissue is an endocrine organ that can release pro-inflammatory cytokines and adipokines that may have a negative impact on respiratory physiology [42].

### 4.4. Elevated REE in Asthma: Contributing Factors

Changes in mitochondrial bioenergetics [4,5] have recently been implicated in asthma pathogenesis. We [4] as well as others [5] have reported that airway epithelial cells and platelets derived from patients with asthma have increased mitochondrial respiration with higher extracellular acidification rates, suggesting increased mitochondrial oxidation in individuals with asthma. We expand upon these findings and provide a clinical-translational link between cellular mitochondrial bioenergetics and whole-body substrate respiration by assessing the respiratory quotient (RQ). Through indirect calorimetry, we obtained the respiratory exchange ratio, which is a ratio of the volume of exhaled CO_2_ to inspired O_2_ and, although it is obtained as a whole-body average of fuel utilization (e.g., carbohydrates and lipids), it accurately represents substrate oxidation at the cellular level. Here, we show that a preference for carbohydrate oxidation (higher RQ) in the fasted state was associated with reduced respiratory function assessed by the FEV_1_/FVC ratio, a relationship that became more pronounced when assessing only the asthmatics. The pulmonary physiology of this relationship can be explained by the increased CO_2_ production derived from carbohydrate oxidation, which requires expulsion via the pulmonary system [43]. This is in agreement with our prior mechanistic work in airway epithelial cells and platelets evidencing glycolytic-shifted mitochondrial bioenergetics (i.e., carbohydrate-shifted fuel preference).

The finding of higher REE in asthma may be related to a greater number of immune cells, in association with upregulation of inflammatory response in asthma [44]. The observation of elevated immune cells in asthma was an anticipated finding, as the inflammatory pathology of asthma has been previously described [45]. Maintenance of immune function accounts for approximately 15% of daily energy expenditure [46]. Energy expended by immune cells are used to support immune cell functions, such as motor functions (migration, cytokinesis, and phagocytosis), antigen processing and presentation, activation functions (signaling), and effector functions (antibody production, cytotoxicity, and regulatory functions) [46]. An increased inflammatory response stimulates energy demand, leading to a metabolic switch of immune cells from energy-efficient oxidative phosphorylation to less efficient aerobic glycolysis for energy production [47,48]. Asthma is classically a Th2-driven inflammatory disease; however, over the past decade, other T cell subsets, such as Th1 and Th17, have also been identified in the mechanisms underlying asthma development [49]. Naïve T cells differentiate into these subsets, and the differentiation results in changes in metabolic requirements and nutrient utilization. Th1, Th2, and Th17 subsets predominantly use aerobic glycolysis, as they require glutamine to support anabolic and rapid proliferation. In agreement, we reported that asthmatics had elevated REE that positively correlated with immune cells, while RQ correlated with lower FEV_1_/FVC and greater bronchodilator reversibility. This suggests that individuals with asthma have greater energy needs and greater carbohydrate use for energy production than healthy controls. This greater carbohydrate nutrient use and energy expenditure may be due to proliferation and/or differentiation of the immune cells that drive inflammation in asthma.

The elevated REE in asthma may also be accounted for, in part, by inherent differences in individuals with asthma, such as increased energy cost of breathing [50] and asthma medications [51]. The increased energy cost of breathing results from abnormal lung mechanics due to airway obstruction [50,52]. However, the asthmatics in this study were mild, well-controlled, with relatively normal pulmonary function tests. Asthma medications, such as β2-adrenergic agonists [53,54,55] and bronchodilator theophylline, are commonly prescribed for asthma symptom management and have been reported to increase REE. In this study, we asked participants to withhold β2-adrenergic agonists medications prior to testing, and no participants were taking theophylline. However, our assessments may still not accurately depict the physiology of medicated adults with asthma. Further, although we implemented a pharmacological washout period that increased with medication half-life, it remains possible that asthma medications may have contributed to the increased REE, particularly the β2-adrenergic receptor agonists and inhaled steroids that are consumed by 71% and 58% of our asthmatic population, respectively [56]. Analysis of REE among asthmatics who use β2-adrenergic receptor agonists or inhaled steroids compared to asthmatics who do not, demonstrated that these medications did not impact REE in this study.

Asthmatics had a higher REE and reported a lower energy intake compared to controls despite similar BMI and body composition, suggesting other factors are contributing to the energy balance equation. It is possible that disrupted mitochondrial bioenergetics play a role as discussed above, but another explanation is differences in physical activity and thermogenesis. We did not assess physical activity or thermogenesis in this study, but these measures will need to be obtained in future research. Physical activity may have been different between groups and has been reported to be reduced in individuals with asthma [57]. Others have reported that reduced fitness (a physiologic measure of volume and intensity of physical activity) and increased sedentary time have independent effects on asthma severity and contribute to reduced total daily energy expenditure [58]. Other components that may impact energy balance were not accounted for in this study, such as diet-induced thermogenesis and non-exercise activity thermogenesis. Based on the available data, it can be deduced that asthmatics have reduced activity compared to controls; however, additional research utilizing doubly labeled water, participant oversight on metabolic wards, and food provision through metabolic kitchens are warranted to more adequately assess whole-body energy balance in asthma.

A substantial body of literature describes the existence of sex-dependent differences in asthma presentation and potential etiology [57]. Although this study was not designed to assess sex differences, the greater number of women in the asthma group may be confounding. We conducted female-only analysis for the outcome measures detailed in this report, which revealed unremarkable differences, primarily reducing correlation coefficients and statistical power, likely due to reductions in the number of participants.

### 4.5. Dietary Intake

The Healthy Eating Index score in our sample was similar to average scores for adults in the United States [58], although asthmatics had lower self-reported energy and carbohydrate intake than healthy controls. Lower reported intake of calories and carbohydrates in asthmatics could be due to a more sedentary lifestyle or a greater magnitude of underreporting, which were not directly assessed in the current study. A more sedentary lifestyle is often accompanied by less activity-related energy expenditure and less dietary compensation to meet activity demands. Thus, lower self-reported calories and carbohydrates may reflect total daily energy expenditure. Other factors, such as age, gender and BMI, also impact under-reporting, although these relationships have not been investigated in asthmatics. Another possibility of lower dietary intake in asthma might relate to olfactory and gustatory functions. Alterations in perceptions of ofactory or gustatory stimuli may negatively affect eating behavior, such as food preferences or the desire to eat, leading to reduced intake of various macro- and micronutrients, and eventually lower overall energy intake [59,60,61]. Olfactory and gustatory functions are tightly linked. Patients with loss of olfactory function report eating less, adding more spices to food, and less preference for sweetness [62]. Recently, Aries-Guillen et al. reported that children with asthma had changes in taste and eating behaviors compared to controls [63], with asthmatic children requiring more time and a higher number of masticatory cycles to finish food, higher concentration to perceive taste, and higher frequency of feeding difficulties. This may be linked to the known impaired sense of smell and altered taste perception in asthmatic individuals [64,65,66,67] related in part to a decrease in expression of olfactory and taste receptors in asthma [68,69]. Intriguingly, olfactory and taste receptors are also found in non-sensory tissues and cells, including the lung [70] and gut [71,72]. In the lung, olfactory and taste receptors are linked to bronchodilation of airway smooth muscle cells and have greater potency than α-adrenergic medications [73].

### 4.6. Limitations

Our study participants were non-severe asthmatics and so the findings might not be applicable to severe asthmatics. Both the asthma and control cohorts were overweight by BMI, but our findings may not translate to more obese cohorts. Although the groups were physiologically similar in BMI, there was a trend toward a higher BMI in the asthma cohort. Dietary intake was assessed via 3-day diet record, a self-report method. Future studies using more rigorous methods to assess energy balance such as doubly-labeled water, inpatient feeding trials, and metabolic chambers are needed to corroborate our findings and provide a more precise quantification of the impact of asthma on energy balance. Finally, this study was performed on a relatively small number of participants, and our results should be interpreted in that context.

## 5. Conclusions

Taken together, our data show that adults with asthma have elevated resting energy expenditure compared to healthy individuals and report consuming less calories and carbohydrates, which suggests an intriguing metabolic paradox in asthma that warrants further investigation.

## Figures and Tables

**Figure 1 nutrients-13-01065-f001:**
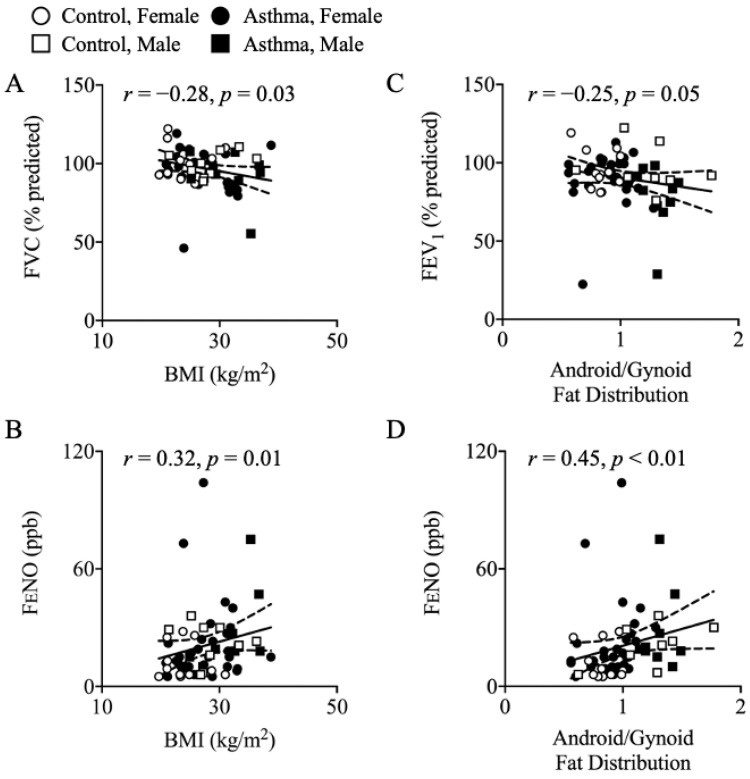
Correlations between Obesity and Body Fat Distribution with Respiratory Function. Obesity and a more android body fat distribution are associated with reduced respiratory function and FENO. BMI was associated with (**A**) lower FVC (percentage-predicted FVC, *r* = −0.28, *p* = 0.03) and (**B**) higher FENO (*r* = 0.32, *p* = 0.01). A higher android-to-gynoid ratio was associated with (**C**) lower FEV_1_ (percentage-predicted FEV_1_, *r* = −0.25, *p* = 0.05) and (**D**) a higher FENO (ppb, *r* = 0.45, *p* < 0.01). FVC, forced vital capacity; FEV_1_, forced expiratory volume in 1 s; FENO, fractional exhaled nitric oxide. Body fat distribution from one participant was unavailable due to not performing the DEXA procedure (control, male).

**Figure 2 nutrients-13-01065-f002:**
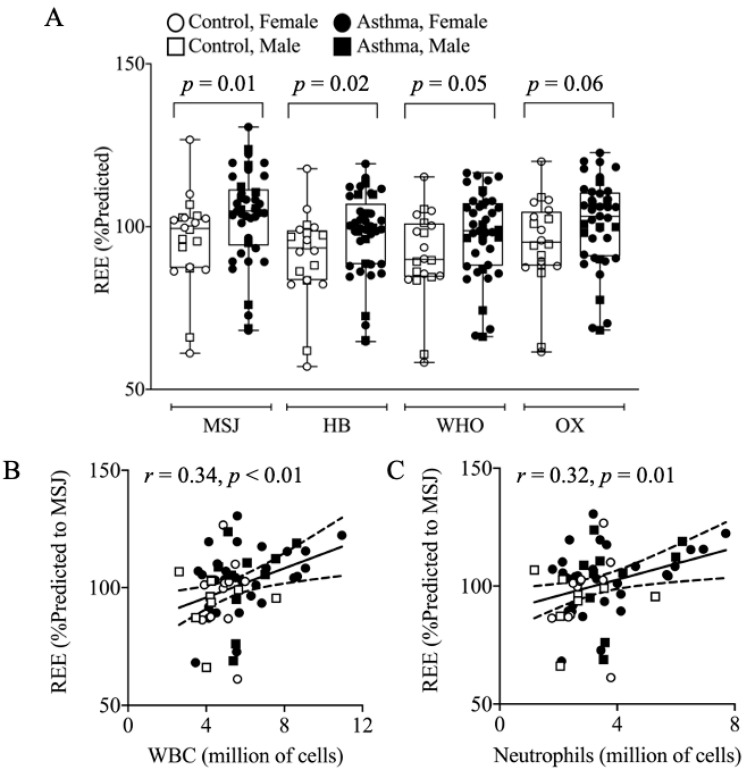
Differences in Resting Energy Expenditure and Correlations with Immune Cells. (**A**) Asthmatic have elevated REE compared to controls, presented as percentage-predicted REE using four validated REE predicted equations. REE (percentage-predicted to MSJ) positively correlated with (**B**) WBCs (*r* = 0.34, *p* < 0.01) and (**C**) Neutrophils (*r* = 0.32, *p* = 0.01). Relationships were similar for HB, WHO, and OX prediction equations. REE, resting energy expenditure; MSJ, Mifflin–St. Jeor; HB, Harris–Benedict; WHO, World Health Organization; OX, Oxford equations; percentage-predicted REE is the percentage of REE_IC_ to REE_MSJ/HB/WHO/OX_.

**Figure 3 nutrients-13-01065-f003:**
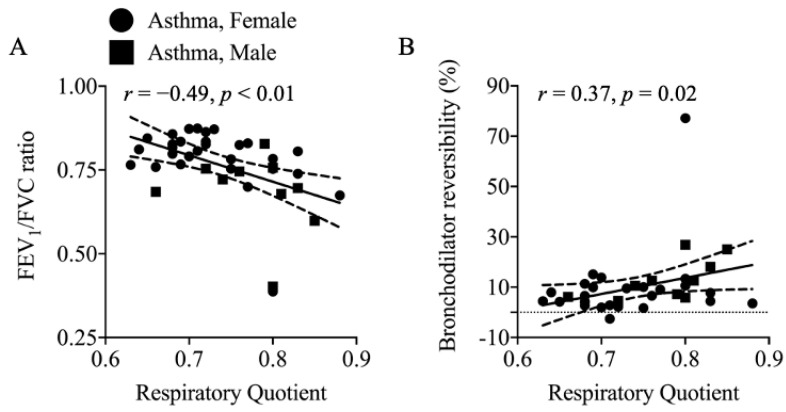
Correlations between Fasting Substrate Oxidation and Respiratory Function in Asthma. A greater propensity to oxidize carbohydrate fuels during fasting metabolism was related to reduced respiratory function. (**A**) RQ was negatively associated with FEV_1_/FVC ratio in adults with asthma (*r* = −0.49, *p* < 0.01). (**B**) RQ was also associated with a more pronounced bronchodilator reversibility (*r* = 0.37, *p* = 0.02). RQ, respiratory quotient, VO_2_/VCO_2_, ratio of the volume of CO_2_ expired over volume of O_2_ inspired during indirect calorimetry, 1.0 = carbohydrate oxidation, 0.7 = fat oxidation; FVC, forced vital capacity; FEV_1_, forced expiratory volume in 1 s; Bronchodilator reversibility, percent change in maximal achievable FEV_1_ after completion of the bronchodilator-treated spirometry compared to baseline.

**Table 1 nutrients-13-01065-t001:** Characteristics of Participants.

	Control	Asthma	*p*-Value
*n* (sex: male, female)	20 (9 m, 11 f)	41 (10 m, 31 f)	-
Age (years)	37.5 ± 11.3	38.9 ± 11.0	0.64
SBP (mmHg)	123.8 ± 15.0	120.4 ± 16.6	0.44
DBP (mmHg)	73.4 ± 8.8	74.8 ± 10.6	0.62
Resting Heart Rate (bpm)	70.1 ± 13.2	71.5 ± 11.3	0.68
Temperature (°F)	97.8 ± 0.5	97.9 ± 0.4	0.71
O_2_ saturation (%)	99.1 ± 1.1	98.9 ± 1.8	0.57
FVC (L)	4.4 ± 0.9	3.8 ± 1.1	0.03
FVC (% predicted)	102.0 ± 10.9	94.3 ± 13.8	0.05
FEV_1_ (L)	3.5 ± 0.7	2.9 ± 0.9	0.02
FEV_1_ (% predicted)	97.1 ± 12.9	87.8 ± 17.6	0.08
FEV_1_/FVC ratio	0.79 ± 0.07	0.76 ± 0.10	0.45
FENO (ppb)	16.8 ± 10.7	26.3 ± 30.2	0.26

Data represent mean ± SD. SBP, systolic blood pressure; DBP, diastolic blood pressure; O_2_ saturation, oxygen saturated hemoglobin relative to total hemoglobin; FVC, forced vital capacity; FEV_1_, forced expiratory volume in 1 s; FENO, fractional exhaled nitric oxide. Statistical significance determined by unpaired Student’s *t*-test or Mann–Whitney *U* test when data were not normally distributed.

**Table 2 nutrients-13-01065-t002:** Body Composition.

	Control			Asthma			*p*-Value		
	**Cohort**	**Male** ***n* = 9**	**Female** ***n* = 11**	**Cohort**	**Male** ***n*** **= 10**	**Female** ***n*** **= 31**	**Cohort**	**Male**	**Female**
Height (cm)	171.5 ± 9.9	179.2 ± 8.5	165.2 ± 5.5	168.6 ± 8.7	178.1 ± 8.6	165.6 ± 6.2	0.25	0.78	0.85
Weight (kg)	77.1 ± 17.7	91.3 ± 14.4	65.4 ± 10.0	81.0 ± 15.9	99.0 ± 12.1	75.2 ± 12.2	0.26	0.22	0.02
BMI (kg/m^2^)	26.0 ± 4.4	28.4 ± 4.4	24.0 ± 3.4	28.4 ± 4.7	31.3 ± 4.5	27.4 ± 4.4	0.06	0.17	0.02
**DEXA Data**	**Cohort**	**Male ^†^** ***n* = 8**	**Female** ***n* = 11**	**Cohort**	**Male** ***n*** **= 10**	**Female** ***n*** **= 31**	**Cohort**	**Male**	**Female**
BMD (g/cm^2^)	1.21 ± 0.18	1.33 ± 0.10	1.13 ± 0.18	1.25 ± 0.11	1.36 ± 0.11	1.21 ± 0.09	0.36	0.53	0.05
Fat mass (%)	33.9±8.3	30.1±9.2	36.6±6.7	38.7±7.5	33.7±7.8	40.3 ± 6.7	0.03	0.39	0.12
Fat mass (kg)	24.6 ± 8.5	26.4 ± 10.1	23.3 ± 7.3	30.4 ± 9.5	32.5 ± 10.1	29.8 ± 9.3	0.02	0.22	0.04
Fat-free mass (kg)	48.2 ± 13	60.1 ± 11.0	39.5 ± 4.8	47.5 ± 10.2	62.8 ± 8.2	42.6 ± 4.1	0.83	0.55	0.05
FFMI (kg/m^2^)	16.3 ± 2.9	18.8 ± 2.6	14.5 ± 1.3	16.6 ± 2.6	19.9 ± 2.8	15.5 ± 1.4	0.67	0.41	0.03
Gynoid fat (%)	36.4 ± 9.4	29.6 ± 8.0	41.3 ± 7.0	41.8 ± 8.3	32.5 ± 7.3	44.8 ± 6.2	0.03	0.43	0.13
Android fat (%)	36.1 ± 12.0	37.6 ± 15.2	35.1 ± 9.7	41.5 ± 11.0	42.7 ± 10.3	41.1 ± 11.3	0.09	0.41	0.12
A/G Ratio	1.00 ± 0.3	1.2 ± 0.3	0.8 ± 0.1	1.0 ± 0.2	1.3 ± 0.1	0.9 ± 0.2	0.94	0.46	0.23

Data represent mean ± SD. DEXA, dual-energy X-ray absorptiometry; BMI, body mass index; BMD, bone mineral density; FFMI, fat-free mass index, fat-free mass (kg)/height (m)^2^; A/G Ratio, android-to-gynoid fat distribution ratio. Statistical significance determined by unpaired Student’s *t*-test or Mann–Whitney *U* test when data were not normally distributed. ^†^ Body composition data were unavailable for one participant who did not undergo the DEXA procedure due to clinical concerns, and data are not present in this analysis (control, male).

**Table 3 nutrients-13-01065-t003:** Energy Expenditure.

	Control			Asthma			*p*-Value		
Indirect Calorimetry	Cohort	Male *n* = 9	Female *n* = 11	Cohort	Male *n* = 10	Female *n* = 31	Cohort	Male	Female
REE (kcals/day)	1505 ± 342	1738 ± 289	1314 ± 258	1594 ± 337	1954 ± 398	1477 ± 216	0.34	0.20	0.05
RQ	0.77 ± 0.10	0.77 ± 0.08	0.78 ± 0.11	0.74 ± 0.06	0.78 ± 0.06	0.73 ± 0.06	0.18	0.88	0.13
Prediction Equations									
MSJ (kcals/day)	1572 ± 284	1838 ± 168	1355 ± 124	1552 ± 249	1905 ± 149	1438 ± 144	0.97	0.60	0.07
Residuals	67 ± 221	100 ± 231	40 ± 220	−42 ± 236	−49 ± 351	−40 ± 194	0.02	0.09	0.10
Residuals (%)	7.0 ± 19.6	7.8 ± 17.5	6.4 ± 22.0	−0.6 ± 16.5	0.8 ± 21.1	−1.1 ± 15.0	0.01	0.09	0.09
HB (kcals/day)	1662 ± 306	1943 ± 216	1432 ± 105	1639 ± 270	2040 ± 192	1510 ± 126	0.80	0.45	0.04
Residuals	157 ± 229	205 ± 243	118 ± 221	46 ± 228	85 ± 333	33 ± 188	0.01	0.16	0.14
Residuals (%)	13.2 ± 21.3	13.8 ± 19.3	12.7 ± 23.8	4.9 ± 16.9	7.6 ± 21.7	4.0 ± 15.4	0.02	0.16	0.13
WHO (kcals/day)	1659 ± 302	1960 ± 153	1413 ± 82	1641 ± 296	2080 ± 168	1500 ± 154	0.81	0.24	0.03
Residuals	154 ± 238	222 ± 265	99 ± 208	48 ± 222	126 ± 290	22 ± 195	0.06	0.45	0.23
Residuals (%)	13.0 ± 21.2	15.3 ± 20.5	11.2 ± 22.5	4.7 ± 16.6	9.5 ± 19.2	3.2 ± 15.7	<0.05	0.40	0.16
OX (kcals/day)	1594 ± 307	1890 ± 186	1352 ± 92	1586 ± 297	2021 ± 192	1445 ± 151	0.80	0.28	0.03
Residuals	89 ± 233	152 ± 261	38 ± 205	-8 ± 220	66 ± 283	−32 ± 195	0.07	0.50	0.22
Residuals (%)	8.4 ± 20.0	11.0 ± 19.6	6.3 ± 21.1	1.1 ± 16.1	6.2 ± 18.2	−0.6 ± 15.3	0.06	0.50	0.18

Data represent mean ± SD. REE, resting energy expenditure; IC, indirect calorimetry; RQ, respiratory quotient, VO_2_/VCO_2_; FFM, fat-free mass; MSJ, Mifflin–St. Jeor; HB, Harris–Benedict; WHO, World Health Organization equations; OX, Oxford equations; Residuals, difference between predicted-REE, and measured-REE (residuals = REE_MSJ/HB/WHO/OX_ − REE_IC_), kcals/day. Statistical significance determined by unpaired Student’s *t*-test or Mann–Whitney *U* test when data were not normally distributed.

**Table 4 nutrients-13-01065-t004:** Prediction Equation Adjustments for Asthma Status and FENO.

Model	Adjusted *R*^2^	ANOVA *p*-Value	Coefficient	*p*-Value
MSJ	0.511	<0.01		
MSJ + Asthma	0.526	<0.01	107.8	0.10
MSJ + FENO	0.527	<0.01	2.1	0.09
HB	0.529	<0.01		
HB + Asthma	0.545	<0.01	108.8	0.09
HB + FENO	0.551	<0.01	2.2	0.06
WHO	0.544	<0.01		
WHO + Asthma	0.558	<0.01	103.8	0.10
WHO + FENO	0.550	<0.01	1.6	0.18
OX	0.560	<0.01		
OX + Asthma	0.571	<0.01	95.8	0.12
OX + FENO	0.569	<0.01	1.7	0.14

MSJ, Mifflin–St. Jeor; HB, Harris–Benedict; WHO, World Health Organization; OX, Oxford equations; FENO, fractional exhaled nitric oxide.

**Table 5 nutrients-13-01065-t005:** Summary Statistics of REE Linear Regression Models.

Model	Adjusted *R*^2^	ANOVA *p*-Value	Coefficient	*p*-Value
Model 1 Age, Sex, Fat mass, and Fat-free Mass	0.552	<0.0001		
Model 2 Model 1 + Asthma	0.554	<0.0001	73.9	0.28
Model 3 Model 1 + FENO (ppm)	0.563	<0.0001	1.9	0.13
Model 4 Model 1 + White blood cell count (×10^6^ cells)	0.588	<0.0001	45.3	0.02
Model 5				
Model 1 + Neutrophil count (×10^6^ cells)	0.580	<0.0001	48.3	0.04

Cohort-specific REE equations with multiple linear regression modeling. Model 1 was created with age (years), sex (male = 1, female = 0), fat mass (kg), and fat-free mass (kg) as variables. Models 2, 3, 4, and 5 were created by adding asthma status (asthma diagnosis = 1, no asthma diagnosis = 0), FENO (ppm), white blood cell or neutrophil counts (millions of cells) to Model 1, respectively. Observations, *n* = 60 (1 participant was excluded due to not performing a DEXA). Detailed summary statistics are available in Appendix A.

**Table 6 nutrients-13-01065-t006:** Blood Chemistry and Immune Cells.

	Control *n* = 20	Asthma *n* = 41	*p*-Value
WBC	4.7 ± 1.1	5.9 ± 1.6	<0.01
RBC	4.7 ± 0.4	4.8 ± 0.6	0.51
Hemoglobin	13.7 ± 1.4	13.9 ± 1.5	0.55
Hematocrit	41.7 ± 3.7	42.5 ± 4.2	0.47
Platelets	240 ± 43	242 ± 60	0.93
MPV	8.6 ± 0.7	8.5 ± 0.9	0.72
Neutrophil (%)	58.1 ± 9.0	60.6 ± 8.5	0.30
Neutrophil (millions of cells)	2.8 ± 0.9	3.6 ± 1.4	0.02
Lymphocyte (%)	31.2 ± 7.8	28.6 ± 7.2	0.21
Lymphocyte (millions of cells)	1.4 ± 0.4	1.6 ± 0.4	0.11
Monocyte (%)	5.2 ± 1.3	4.9 ± 1.2	0.33
Monocyte (millions of cells)	0.2 ± 0.1	0.3 ± 0.1	0.10
Eosinophil (%)	2.5 ± 1.7	3.1 ± 3.0	0.36
Eosinophil (millions of cells)	0.1 ± 0.1	0.2 ± 0.2	0.14
Basophil (%)	0.6 ± 0.2	0.7 ± 0.4	0.60
Basophil (millions of cells)	0.03 ± 0.01	0.04 ± 0.02	0.16

Data represent mean ± SD. WBC, white blood cells; RBC, red blood cells; MPV, mean plasma volume. Statistical significance determined by Student’s *t*-test or Mann–Whitney *U* test when data were not normally distributed.

**Table 7 nutrients-13-01065-t007:** Nutritional Analysis.

	Control			Asthma			*p*-Value		
	Cohort	Male *n* = 9	Female *n* = 11	Cohort	Male *n* = 10	Female ^†^ *n* = 29	Cohort	Male	Female
Calorie Intake (kcals/day)	1927 ± 751	2337 ± 901	1591 ± 381	1603 ± 411	1805 ± 474	1534 ± 371	0.10	0.20	0.60
Calorie Intake (kcals/kg/day)	25.1 ± 7.5	25.4 ± 8.4	24.9 ± 7.1	20.3 ± 6.0	18.8 ± 6.5	20.8 ± 5.9	<0.01	0.07	0.07
Protein (g/day)	85 ± 47	110±61	65 ± 30	73 ± 31	87 ± 34	68 ± 29	0.20	0.28	0.68
Protein (g/kg/day)	1.1 ± 0.4	1.2 ± 0.6	1.0 ± 0.3	0.9 ± 0.4	0.9 ± 0.4	0.9 ± 0.4	0.07	0.24	0.23
Fat (g/day)	83 ± 42	105 ± 54	65 ± 17	70 ± 24	74 ± 18	69 ± 26	0.51	0.21	0.77
Fat (g/kg/day)	1.1 ± 0.4	1.1 ± 0.5	1.0 ± 0.3	0.9 ± 0.4	0.8 ± 0.2	0.9 ± 0.4	0.10	0.06	0.57
Saturated fat (g/day)	25 ± 10	31 ± 12	21 ± 5	23 ± 9	24 ± 8	23 ± 9	0.33	0.16	0.56
Carbohydrates (g/day)	208 ± 69	236 ± 62	184 ± 68	161 ± 53	191 ± 65	151 ± 46	<0.01	0.14	0.08
Carbohydrates (g/kg/day)	2.8 ± 1.0	2.6 ± 0.6	2.9 ± 1.2	2.0 ± 0.7	2.0 ± 0.8	2.0 ± 0.7	<0.01	0.09	<0.01
Sugar (g/day)	83 ± 37	77 ± 38	87 ± 38	57 ± 28	69 ± 34	53 ± 24	<0.01	0.63	<0.01
Fiber (g/day)	16 ± 8	16 ± 4	16 ± 11	15 ± 7	16 ± 11	15 ± 5	0.64	0.37	0.93
HEI (total score)	57 ± 11	51 ± 9	62 ± 11	54 ± 12	48 ± 10	55 ± 12	0.29	0.61	0.10

Data represent mean ± SD. Means represent daily average intake obtained from 3-day dietary records. HEI, Healthy Eating Index score; Statistical significance determined by unpaired Student’s *t*-test or Mann–Whitney U test when data were not normally distributed. ^†^ Two participants did not provide 3-day dietary records and are not present in this analysis (*n* = 2, asthmatic, female).

## Data Availability

The data presented in this study are available upon request from the corresponding author.

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
