# Peer review of "Resting Energy Expenditure Is Elevated in Asthma"

_nutrients, 2021, doi:10.3390/nu13041065_

Round 1
Reviewer 1 Report
In this manuscript the authors investigated resting energy expenditure in patients with asthma, in comparison with healthy controls. The results are very interesting and the study is conducted with scientific rigor.
Comments:
- the authors should better explain the possible interferences of pharmacological therapy, suspended only 24 hours before
- the authors should better argue the relationship with blood cells, in fact it is well known that asthma is an inflammatory pathology, therefore the increase of cells in the blood count was expected
- as the authors correctly argue in the limitations, it was important to know at least at baseline the physical activity status of the patients and subjects studied
Author Response
Dear Reviewers,
We would like to say thank you for the opportunity to submit a revised draft of our manuscript title, "Resting Energy Expenditure is Elevated in Asthma". We appreciate the time and efforts that the reviewers have dedicated to review our manuscript thoughtfully. We are grateful to the reviewers for their insightful comments and valuable feedback to improve our manuscript. We have carefully taken these comments into consideration in revising this manuscript. We have incorporated changes to reflect most of the suggestions provided and we hope that the manuscript is now suitable for publication.
Below is a point-by-point response to the reviewers' comments and concerns.
Response to Reviewer #1 Comments:
Comment 1:
The authors should better explain the possible interferences of pharmacological therapy, suspended only 24 hours before.
Response to Comment 1:
Thank you to the reviewer for pointing out the potential impact of asthma medication in this study. We agree with reviewer #1 that pharmacological therapy can contribute to positive or negative changes in REE, as reported in many studies. We have anticipated the contribution of these medications on our REE data. Therefore, we have collected and reported each participant’s medication information, provided in Supplementary Table S4. Further, we also analyzed the contribution of these medications on REE, mainly concerned about the contribution of short acting beta agonists and inhaled corticosteroids; these data are provided in Supplementary Table S9 and Table S10. In our population, we did not find any significant differences in REE between asthmatic subjects who took short acting beta agonists or inhaled corticosteroids compared to those who did not take those medications. We also did not see the effect of leukotriene modifiers on REE in our population (data not shown). We could not assess the impact of oral corticosteroid, biologics or other short- or long-acting muscarinic antagonist on REE in current study, due to a limited number of participants using those medications (less than 10% of asthmatic population). Towards the reviewer’s point, we have expanded our commentary on this topic in our discussion. (Discussion, at part 4.4. Elevated REE in Asthma: Contributing Factors, line 377 - 384).
Comment 2:
The authors should better argue the relationship with blood cells, in fact it is well known that asthma is an inflammatory pathology, therefore the increase of cells in the blood count was expected.
Response to Comment 2:
Thank you for this comment on asthma physiology. We have expanded our discussion on this topic at part 4.4 Elevated REE in Asthma: Contributing Factors, line 342-354.
Comment 3:
As the authors correctly argue in the limitations, it was important to know at least at baseline the physical activity status of the patients and subjects studied.
Response to Comment 3:
We agree with the reviewer that baseline physical status of the subjects is an important factor on REE. We did not collect any baseline physical activity data of our participants. We admit that this is a limitation in our current study, as the original objective was purely to assess the relationship of metabolism to asthma status, not on their physical activity or exercise.
Reviewer 2 Report
The thematic is interesting, potential impact in clinical care. To maintain adequate nutritional status is important to access energy expenditure. Since indirect calorimetry is not yet available in routine care, knowledge that individuals with asthma have higher REE than expected, allow tailor the nutritional intervention to promote adequate nutritional status.
152 (Table 2) - Since FFM is the major responsible for REE, it will be interesting not only present the crud results, but to classify the FFM (example as FFM index) of subjects.
305 - Authors found that independent of asthma status, an android-type fat distribution were related to reduce respiratory function. – Suggest present a possible explanation for this findings.
324 – I agree that expulsion, via the pulmonary system, of the increased CO2 production derived from carbohydrate oxidation, will be more difficult in asthma patients, explaining the higher RQ in asthma sample. Authors also reinforce data from a previous study, evidencing glycolytic-shifted mitochondrial bioenergetics (i.e., carbohydrate-shifted fuel preference) in airway epithelial cells and platelets. But fuel preferences also depend on the type of diet. A ketogenic diet will shift the fuel preferences. In asthma woman sample this could had happen, observing the mean values, with intake of 1534±371 Kcal/day, carbohydrates intake of 151±41g/day and fat intake of 69±26g/day, these will correspond to a mean percentage of energy from carbohydrates and from fat around 40% each, even a little higher from fats. Will be interesting to know if there are individuals with a ketogenic diet and if this is also responsible for the RQ findings.
Author Response
Dear Reviewers,
We would like to say thank you for the opportunity to submit a revised draft of our manuscript title, "Resting Energy Expenditure is Elevated in Asthma". We appreciate the time and efforts that the reviewers have dedicated to review our manuscript thoughtfully. We are grateful to the reviewers for their insightful comments and valuable feedback to improve our manuscript. We have carefully taken these comments into consideration in revising this manuscript. We have incorporated changes to reflect most of the suggestions provided and we hope that the manuscript is now suitable for publication.
Below is a point-by-point response to the reviewers' comments and concerns.
Response to Reviewer #2 Comments:
Comment 1:
152 (Table 2) - Since FFM is the major responsible for REE, it will be interesting not only present the crud results, but to classify the FFM (example as FFM index) of subjects.
Response to Comment 1:
Thank you to the reviewer. We have calculated and added the fat-free mass index (FFMI) data on Table 2 for the study participants. The FFMI for Asthma is 16.6±2.6 kg/m2 and Control 16.3±2.9 kg/m2, p=0.67 between groups. This is anticipated because without a physiology that reduces FFM (like cachexia or sarcopenia) or a lifestyle and diet that increases FFM (athlete, resistance training in combination with a high protein diet), FFMI should be similar among participants. Out of additional interest to this reviewer, we conducted correlational analysis of FFMI with our primary lung function and asthma outcome measures (% predicted FEV1, p=0.26; % predicted FVC, p=0.50; FEV1/FVC, p=0.13; bronchodilator reversibility p=0.37; FENO, p=0.20) and found no correlation of FFMI with these outcomes. Due to these non-significant findings and the lack of an a priori hypothesis on our end, we have decided to not report these data in the main manuscript but provide them here for your reference.
Comment 2:
305 - Authors found that independent of asthma status, an android-type fat distribution was related to reduce respiratory function. – Suggest present a possible explanation for this finding.
Response to Comment 2:
Thank you for pointing this out. We have added these possible explanations in our discussion (Discussion, section 4.3 Body Composition, line 315 - 322).
Comment 3:
324 – I agree that expulsion, via the pulmonary system, of the increased CO2 production derived from carbohydrate oxidation, will be more difficult in asthma patients, explaining the higher RQ in asthma sample. Authors also reinforce data from a previous study, evidencing glycolytic-shifted mitochondrial bioenergetics (i.e., carbohydrate-shifted fuel preference) in airway epithelial cells and platelets. But fuel preferences also depend on the type of diet. A ketogenic diet will shift the fuel preferences. In asthma woman sample this could had happen, observing the mean values, with intake of 1534±371 Kcal/day, carbohydrates intake of 151±41g/day and fat intake of 69±26g/day, these will correspond to a mean percentage of energy from carbohydrates and from fat around 40% each, even a little higher from fats. Will be interesting to know if there are individuals with a ketogenic diet and if this is also responsible for the RQ findings.
Response to Comment 3:
We agree with this reviewer that fuel preferences are impacted by diet. Notably, we implemented an overnight fast prior to RQ testing, so small differences in macronutrient composition during a weight-stable lifestyle are unlikely to impact RQ. However, to this reviewer’s point, a diet that induces systemic changes in fuel utilization, like a ketogenic diet, may have effects that extend beyond our overnight fast control procedure. To respond to this point, we calculated the % of energy coming from carbohydrates (CHO), which was 41.7% for the whole group. This was slightly lower, but not statistically different in the asthma group (41% Asthma vs 44% control). The lowest habitual %CHO reported was 10%, with 63% energy from fat, 22% protein and 5% alcohol. This participant’s RQ was in the lowest tertile of RQs reported, but it was not the lowest. Although we cannot draw conclusions as to whether this person’s diet increased ketogenesis without additional measures, we agree that the topic of ketogenesis and a ketogenic diet in asthma is interesting and requires additional investigation.